# Comparison between five acellular oxidative potential measurement assays performed with detailed chemistry on PM$_{10}$ samples from the city of Chamonix (France)

Aude Calas[1], Gaëlle Uzu[1], Frank J. Kelly[2], Stephan Houdier[1], Jean M.F. Martins[1], Fabrice Thomas[3], Florian Molton[3], Aurélie Charron[1, 4], Christina Dunster[2], Ana Oliete[2], Véronique Jacob[1], Jean-Luc Besombes[5], Florie Chevrier[1, 5], and Jean-Luc Jaffrezo[1]

[1]Univ. Grenoble Alpes, CNRS, IRD, Grenoble INP, IGE, F-38000 Grenoble, France
[2] MRC-PHE Centre for Environment and Health, Environmental Research Group (ERG), King's College London, London, United Kingdom Kingdom
[3]Univ. Grenoble Alpes, DCM, F-38 000 Grenoble, France
[4]IFSTTAR, F-69675 Bron, France.
[5]Univ. Savoie Mont Blanc, LCME, 73000 Chambéry, France

*Correspondence to*: Gaëlle Uzu (gaelle.uzu@ird.fr)

**Abstract.** Many studies have demonstrated associations between exposure to ambient particulate matter (PM) and adverse health outcomes in humans that can be explained by PM capacity to induce oxidative stress in vivo. Thus, assays have been developed to quantify the oxidative potential (OP) of PM as a more refined exposure metric than PM mass alone. Only a small number of studies have compared different acellular OP measurements for a given set of ambient PM samples. Yet, fewer studies have compared different assays over a year long period and with detailed chemical characterization of ambient PM. In this study, we report on seasonal variations of the dithiothreitol (DTT), ascorbic acid (AA), electron spin resonance (ESR) and the respiratory tract lining fluid (RTLF, composed of the reduced glutathione (GSH) and ascorbic acid (ASC)) assays over a one year period in which 100 samples were analysed. A detailed PM$_{10}$ characterization allowed univariate and multivariate regression analyses in order to obtain further insight into groups of chemical species that drive OP measurements. Our results show that most of the OP assays were strongly inter-correlated over the sampling year but also these correlations differed when considering specific sampling periods (cold vs warm). All acellular assays are correlated with a significant number of chemical species when considering univariate correlations, especially for the DTT assay. Evidence is also presented of a seasonal contrast over the sampling period with significantly higher OP values during winter for the DTT, AA, GSH and ASC assays, which were assigned to biomass burning species by the multiple linear regression models. The ESR assay clearly differs from the other tests as it did not show seasonal dynamics and presented weaker correlations with other assays and chemical species.

# 1 Introduction

Many studies have demonstrated associations between exposure to ambient particulate matter (PM) and adverse health outcomes in humans. A central mechanism to explain the harmful effects of a range of inhaled particles at the cellular level involves the production of oxidative stress through the generation of excessive reactive oxygen species (ROS) and/or

inadequate antioxidant defenses (Borm et al., 2007; Kelly, 2003). The capacity of PM to elicit damaging oxidative reactions is termed oxidative potential (OP).

On this basis, probes have been developed over the last decade to quantify the OP of PM as a more refined exposure metric of PM toxicity than PM mass alone (Ayres et al., 2008; Borm et al., 2007). These probes include several acellular assays. The most common consisting in mimicking the consumption of antioxidants (*e.g.* ascorbic acid (AA), reduced glutathione

(GSH)) or surrogates (e.g. dithiothreitol (DTT)), the use of the synthetic human respiratory tract lining fluid (RTLF) system (again to assess antioxidant depletion), probes measuring $HO^{\bullet}$ production or the application of electron spin resonance (ESR) to quantify the ability of PM to induce specific ROS (e.g. $HO^{\bullet}$ radicals). Oxidative potential can be considered as an integrative metric of PM characteristics (size, composition, surface area…) potentially linked to the particles toxicity through oxidative stress. Therefore, they could help to delineate those particle properties (components and sources) responsible for

observed health effects. However, each of these assays is sensitive to different panels of ROS generating compounds, and results are also sensitive to the assay design. Indeed a consensus has yet to emerge regarding a standard in vitro test system or combination of tests that would be most appropriate for PM-related health impact evaluation (Ayres et al., 2008).

Only a small number of studies have compared different acellular OP measurements for a given set of ambient PM samples (Fang et al., 2016; Janssen et al., 2014; Künzli et al., 2006; Szigeti et al., 2015; Visentin et al., 2016; Yang et al., 2014). Yet,

fewer studies have compared different assays over a year long period to gain a better understanding of seasonal variability (Fang et al., 2016; Jedynska et al., 2017; Saffari et al., 2014; Szigeti et al., 2015; Yang et al., 2015). Finally, there is little research relating the oxidative capacity of particulate pollution with detailed chemical characterization of ambient PM, in an attempt to identify the PM components or sources that may contribute most to underlying toxicity (Fang et al., 2016; Kelly et al., 2011; Saffari et al., 2014; Verma et al., 2014; Weber et al., 2018).

In this study, a series of 100 $PM_{10}$ samples collected over a one year period were screened for ROS burden using 4 acellular measures of OP: DTT, AA, ESR and RTLF assays. The RTLF assay includes 3 antioxidants: reduced glutathione (GSH), ascorbic acid (referred as ASC) and urate (UA). Two ascorbic acid depletion assays (AA and ASC), using different quantification techniques, were therefore included in our analyses. DTT is known to react with organic compounds but also with transition metals (Charrier and Anastasio, 2012; Lin and Yu, 2011). The ESR assay employs the spin trap

5,5-Dimethyl-1-pyrroline N-oxide (DMPO) and is specific for reactive radical species, which for example result from partial reduction of dioxygen catalyzed by transition metal (Boogaard et al., 2012; Shi et al., 2003). The AA assays would be more specific to the oxidative potential of transition metals (Godri et al., 2011; Yang et al., 2014), but ascorbic acid is known to react with organics such as quinones (Shang et al., 2012; Visentin et al., 2016). All of these assays have shown some

correlations with health outcomes in epidemiological studies (Abrams et al., 2017; Bates et al., 2015; Fang et al., 2016; Strak et al., 2017; Weichenthal et al., 2016a, 2016b; Yang et al., 2016). Detailed $PM_{10}$ characterization was performed in parallel, by analyzing up to 130 chemical species that incorporated a broad array of organic species and trace elements (Chevrier, 2016).

In this paper, we report on seasonal variations within the redox activity assays as well as on the correlation among the different assays over a one year period. Univariate and multivariate regression analyses were applied in order to obtain (a) further insight into groups of chemical species that drive OP measurements and (b) evaluate if differences could be detected between the individual assays.

## 2 Material and methods

### 2.1 Site description and sampling

In Europe, particulate matter sanitary alert are based on $PM_{10}$ measurements. $PM_{10}$ were collected in the downtown area of Chamonix (45°55'21.53" N, 6°52'11.68" E, Auvergne-Rhône Alpes, France, 1035 masl) in the Alpine Arve valley. This urban location is heavily impacted in winter by biomass burning (wood combustion used for domestic heating) and traffic emissions. Further, because of their topography, specific weather conditions and anthropogenic activities, the European daily

limit value for $PM_{10}$ is often exceeded in many sites in the Alpine valleys during the winter period (Chevrier, 2016), including this site in Chamonix.

In the framework of the DECOMBIO project (biomass burning contribution to $PM_{10}$ in the Arve's Valley) $PM_{10}$ sampling and detailed chemical characterizations have been achieved, and are described elsewhere (Chevrier, 2016). Briefly, ambient particles were collected by filtration during 24 h ($24 \times 30$ $m^3.h^{-1}$) with a DIGITEL DA-80 on 150 mm quartz filters

(Tissuquartz Pallflex) using the European standard protocol NF EN 16450. DIGITEL DA-80 was automatically program to stock before and after sampled filters, and the samples were then collected every week. The filters were calcined at 500 °C for 8 h before use. After sampling, the filters were folded, wrapped in aluminum foils, sealed in polyethylene bags and stored at -25 °C until chemical analyses and at 4 °C until OP analysis. $PM_{10}$ mass measurements were achieved at the sampling site with TEOM-FDMS, as part of the regular Atmo-AURA network of Air Quality observation (http://www.air-rhonealpes.fr/).

### 2.2 Chemical analyses

### 2.2.1 $PM_{10}$ chemical composition

Briefly, collected $PM_{10}$ samples were measured for the following elements and components: elemental and organic carbon (EC ,OC), BC and the distinction between wood burning ($BC_{wb}$) and fossil fuel BC ($BC_{ff}$), soluble anions and cations ($NO_3^-$, $SO_4^{2-}$, $Cl^-$, MSA, oxalate and $NH_4^+$, $Mg^{2+}$, $Na^+$, $Ca^{2+}$, $K^+$), a large range of inorganic elements (Al, Fe, Ti, As, Ba Cd, Ce, Cr,

Cu, La, Li, Mn, Mo, Ni, Pb, Rb, Sb, Sn, Sr, V, Zn and Zr), sugar alcohols (arabitol, sorbitol, and mannitol, also called ∑

Polyols), monosaccharide anhydrides (levoglucosan, mannosan and galactosan, ∑ Monosaccharides), humic like substances (HULIS), and polar and apolar organics tracers (alkanes (∑ alkanes), hopanes (∑ hopanes), methoxyphenols (∑ methoxyphenols), polycyclic aromatic hydrocarbons (∑ PAHs), substituted derivatives (methyl-PAHs) and polycyclic aromatic sulfur heterocycles (∑ PASHs). More detailed information is available in the SI (Section 1).

**2.2.2 Oxidative Potential assays**

A total of 98 $PM_{10}$ samples collected, from November 2013 to October 2014, were analyzed for redox activity. Since studies have shown a non-linear DTT response to both PM concentrations (Charrier et al., 2016) or from different chemical species added to the assay (Calas et al., 2017; Wang et al., 2017), for the DTT and AA assays (single compound assay), extractions were achieved for each sample to a final concentration of 10 µg.ml$^{-1}$ allowing samples inter-comparison as same extractions

at constant-mass were used. For extraction procedure, PM samples were extracted in using a Gamble + DPPC (dipalmitoylphosphatidylcholine) solution and vortexed at maximum speed during 2h at 37°C (Calas et al., 2017). This solution allows for the extraction of PM in an environment closer to physiological conditions. Our previous results (Calas et al., 2017) also show an improvement of PM suspension during the assay thus facilitating the OP DTT measurement. In this study, non-linear response to PM concentrations was observed in the case of the DTT assay but not in the case of the AA

assays (more details can be found in the SI Section 2 and Figure S1). Conversely, for RTLF and ESR assays, no extraction procedure was performed. Filter punches of 0.196 cm² were used in a "direct" measurement (cf. individual subsections for more information).

PM OP measurements with the DTT, AA (single compound assay) and ESR assays were performed at the Grenoble Alpes University in early 2016. The RTLF assay was measured at the King's College of London between late 2014 and 2015.

**2.2.2.1 DTT assay**

A semi-automated procedure was used with a plate-reader TECAN spectrophotometer Infinite® M 200 pro and 96 well CELLSTAR® multiwall plates from Greiner bio-one®, the assay was modified from the DTT assay of Cho et al. (2005). DTT depletion was monitored for 30 min by adopting the following procedure: (1) measurement of the matrix absorbance ($Abs_{mat}$) of particles and substrate at 412 nm (205 µL of phosphate buffer and 40 µL of PM suspension); (2) injection of 12.5

nmol of DTT (50 µL of 0.25 mM DTT solution in phosphate buffer); (3) for each sample (lab blank included) quantification of DTT immediately (t = 0) and after 15 and 30 min of exposure (50 µL of 1 mM 5,5'-dithiobis(2-nitrobenzoic acid) (DTNB) in phosphate buffer) in triplicate. Positive controls consisted in 1,4-naphthoquinone (1,4-NQ) solution and DTT depletion by 1,4-NQ (40 µL of 24.7 µM stock solution) were quantified only once at each measurement time because of the good repeatability between triplicates (Calas et al., 2017).

The rate of DTT loss (nmol.min$^{-1}$) was determined from the slope of the linear regression of calculated nmol of consumed DTT *vs* time. The amount of remaining DTT was calculated following Eq (1):

$$n_{DTT,i} = \frac{Abs_i * n_{DTT,0}}{Abs_{t0}} \qquad (1)$$

where $n_{DTT,i}$ is the amount of DTT (nmol) at t=i, $Abs_i$ is the absorbance at t=i, $n_{DTT,0}$ is the initial amount of DTT (nmol) and $Abs_{t0}$ is the absorbance at t=0.

The linear regression was considered acceptable when $R^2 > 0.90$ and when less than 70% of the initial amount of DTT had been oxidized (Li et al., 2009). Matrix absorbance (Gamble + DPPC solution and PM) was subtracted from the final absorbance and DTT loss in the lab blanks was subtracted from DTT loss of the samples in order to obtain the actual DTT depletion of the samples. Normalization per cubic meter, representative of human exposure, was chosen and OP DTTv (nmol DTT.min$^{-1}$.m$^{-3}$) was calculated by multiplying DTT depletion per µg of PM added in the wells by $PM_{10}$ mass concentration (µg.m$^{-3}$). The measurement quality was estimated by calculating the coefficient of variation (CV %) of positive controls (1,4-NQ). The CV was below 2% (n=13). For the DTT assay, a total of 98 samples were analyzed. Field blanks (6 samples distributed over the sampling year) were also analyzed and observed contamination remained constant over the year (more details can be found in the SI Section 2 and Figure S2).

### 2.2.2.2 AA assay (single compound assay)

A semi-automated procedure using the same plate reader than for the DTT assay was applied using Greiner UV-Star® 96 well plates, this assay was based on the modified assay from Zielinski et al. (1999) and Mudway et al. (2004). Matrix absorbance ($Abs_{mat}$) measurement of PM and substrate at 265 nm was performed (120 µL of Milli-Q water and 80 µL of PM suspension). Then, 24 nmol of AA (100 µL of 0.24 mM AA solution in Milli-Q water) were injected and absorbance was read at 2min and then every 4 min for 30 min. Positive controls (80 µL of 24.7 µM 1,4-NQ solution) were quantified in duplicates. The rate of AA loss (nmol.min$^{-1}$) was determined from the slope of the linear regression of calculated nmol of consumed AA *vs* time. The amount of remaining AA was calculated in the same way as for DTT (Equation 1). The linear regression was considered acceptable when $R^2 > 0.90$. The matrix absorbance was subtracted from the final absorbance and the AA loss in the lab blanks (blank filter) was subtracted from the AA loss of the samples in order to obtain the actual AA depletion of the samples. AAv (nmol AA. min$^{-1}$. m$^{-3}$) were calculated in the same way as for DTT. The measurement quality was estimated calculating the coefficient of variation (CV %) of the positive control (1,4-NQ). The CV was below 2% (n=7). For the AA single compound assay a total of 98 samples were analyzed. Field blanks were also analyzed and no differences with lab blanks were observed (SI Section 2 and Figure S3).

### 2.2.2.3 ESR assay

A modified ESR assay involving no filter extraction was used (Hellack et al., 2014). This alternative method is highly correlated with that involving filter extraction (Hellack et al., 2014; Yang et al., 2014). This approach is based on the generation of HO$^{\bullet}$ radicals in the presence of hydrogen peroxide, via Fenton-type reactions and their trapping by the nitrone

DMPO. Filter punches (0.196 cm²) were placed in 1 mL tubes, and 125 μL Milli-Q water, 125 μL $H_2O_2$ (0.5 M) and 250 μl of DMPO (0.05 M, ESR grade) were added. Tubes were subsequently vortexed for 15 s before being placed in incubation at 37° C for 40 min under agitation (detailed information in SI Section 3, Figure S4). Suspensions were then vortexed again for 15s and 35 μL of the suspension were transferred to a capillary tube to measure the hydroxyl radical ($HO^{\bullet}$) formation

catalyzed by $PM_{10}$. The EPR spectra were recorded on a Bruker EMXplus spectrometer equipped with a high sensitivity resonator operating at 9.44 GHz at 293 K. Preliminary results have shown a strong decrease of ESR response when Gamble + DPPC solution was used instead of Milli-Q water, thus justifying the use of Milli-Q water in this study (SI Section 3 and Figure S5). Filter blanks and a suspension of CuO (2.73 M) were used for background correction and as a positive control respectively. The 26% (n=7) coefficient of variation of the positive control was attributed from changes in the dispersion of

this insoluble form of Cu. Due to limited remaining samples, extractions could not be performed at identical concentrations for all samples. Corrections were applied to the ESR signal when non-linear pattern vs PM mass was observed (SI Section 3). For the ESR assay, a series of 75 samples only was analyzed due to sample availability. OP ESR was expressed in arbitrary units (A.U.) per cubic meter and will be further referred to as OP ESRv.

### 2.2.2.4 RTLF assay

For the RTLF assay, 0.5 mL of synthetic RTLF containing equimolar concentrations (200μM) of ascorbic acid (ASC), urate (UA) and reduced glutathione (GSH) were added to triplicate tubes containing 0.196 cm² filter punch and incubated for 4 hours at 37°C under constant mixing and as preliminary results indicated linear RTLF response to PM concentrations added to the assay (internal report). The synthetic RTLF was prepared in chelex 100 treated HPLC grade water (pH 7). Particle-free controls at 0 and 4 h (C0 and C4), together with negative (carbon black M120) and positive (CRM NIST1648a) controls, and

lab filter blanks were incubated in parallel (Mudway et al., 2004). After 4 h incubation, the micro-tubes were immediately centrifuged at 13 000 rpm for 1 h at 4°C, followed by removal of aliquots into 100 mM phosphate buffer pH 7.5 (for GSH analysis) or 5% *meta*-phosphoric acid (for ASC and UA analysis). All tubes were immediately stored at -70°C. GSH analysis was derived from the total glutathione and oxidized glutathione (GSSG) analysis using a spectrophotometric enzyme-linked DTNB recycling method based on a modified method described by Baker et al. (1990). The ASC and UA analysis used a

reversed-phase high performance liquid chromatography HPLC system (HPLC, Gilson Scientific UK) with electrochemical detection (EG&G, Princeton applied research, model 400) following the procedure described by Iriyama et al. (1984). Field blanks were used for background correction before estimating the consumption of antioxidants. The consumption rate (% OP) of the antioxidants ASC (% OP ASC), UA (% OP UA) and GSH (% OP GSH) were obtained by reference to the 4 h particle-free control (C4). Conversion from % OP to OP per μg of PM and then to OP per cubic meter were also calculated

and will be referred to as OP GSHv and OP ASCv. Results for UA are not presented, since PM did not deplete urate (Mudway et al., 2004; Zielinski et al., 1999). The lab blanks and carbon black negative control displayed minimal (< 5%)

background oxidation. The positive PM control (CRM NIST 1648a) reacted with up to 35 % consumption of the ASC. The same 98 $PM_{10}$ samples as those used in the AA (single compound assay) and the DTT assays were analyzed.

## 3 Data analyses

All statistical analyses were carried out using the R statistical software 3.4.0. Non-parametric Mann-Whitney tests were used
in order to evaluate the statistical significance between the cold and the warm periods. Nonparametric Spearman's rank correlations ($r_s$) were chosen to assess the strength of possible monotonic relationship between the different OP values and the concentration of the different pollutants measured. Linear model (lm) function in R was used for the multiple regressions. P values <0.05 were considered statistically significant.

## 4 Results and discussion

### 4.1 $PM_{10}$ mass concentration and temperature in Chamonix

Figure 1A shows the mass concentration of $PM_{10}$ over the sampling period in Chamonix, taking into account the days with filters analyzed for OP (SI Table S1). Corresponding temperatures are illustrated in Figure 1 B. The data are clearly characterized by two periods, a cold period (late November to late February, n=30) and the warm period (late May to late September, n=29) with average temperatures of 1.4°C and 14.9°C, respectively. $PM_{10}$ concentrations were 2.6 times higher during the cold period (Table 1) (29 ± 14 $\mu g/m^3$ and 10 ± 2 $\mu g/m^3$ for the cold and warm periods, respectively). The significant differences (Figure 1 C) in $PM_{10}$ concentration between these two periods can be explained by different $PM_{10}$ sources, and frequent temperature inversions in this narrow valley in winter. Investigation of the former using a positive matrix factorization (PMF) approach (Chevrier, 2016) indicates that during winter, the dominant emission source is biomass burning (60% of PM mass on average), with 10% due to traffic and about 18% related to secondary inorganic aerosols (SIA). In summer, the main sources are biogenic activity (40%), SIA (35%), and traffic (10%).

**Table 1 : Median ratios between cold (late November to late February) and warm (late May to late September) periods for PM$_{10}$ mass concentration and OP measurements expressed per m$^3$ (OPv) (*** p< 0.001; non parametric Mann Whitney test, [a]1st and 3rd quartiles of the temporal ratio).**

|  | Cold / Warm period [1st , 3rd][a] |
|---|---|
| **PM$_{10}$ mass concentration** | 2.6*** [2.3, 3.7] |
| **OP DTTv** | 2.1*** [1.7, 2.9] |
| **OP AAv** | 7.1*** [6.0, 9.1] |
| **OP ESRv** | 0.8     [0.7, 1.4] |
| **OP GSHv** | 5.0*** [4.6, 5.8] |
| **OP ASCv** | 8.3*** [7.5, 8.7] |

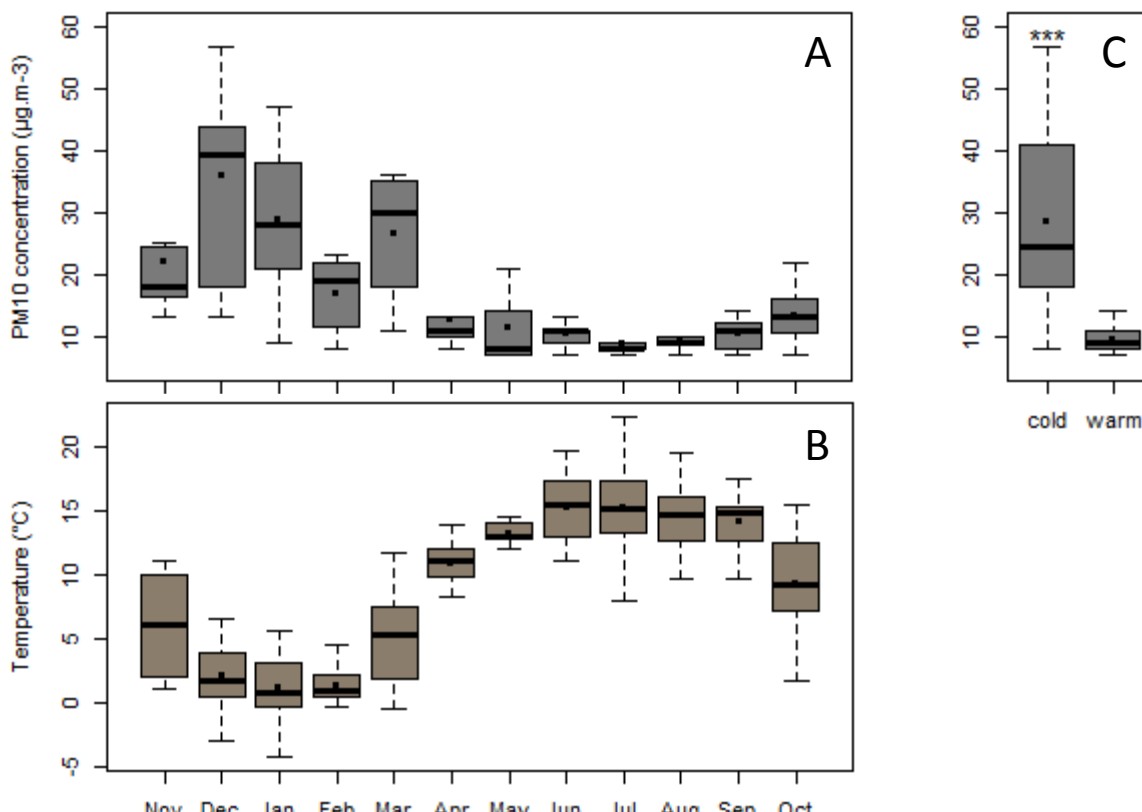

**Figure 1 : Temporal variation in (A) PM10 mass concentration and (B) temperature over the 12 month sampling period. (Average PM10 mass concentration during cold and warm periods (C)). Boxplot representations with medians (horizontal line) and means (black square). *** p <0.001, nonparametric Mann-Whitney test.**

**4.2 OP temporal variation over the 12 month sampling period and during cold or warm periods**

Methodological results on non-linear DTT response to PM concentrations are in agreement with Charrier et al. (2016) and were avoided thank to iso-mass concentration extraction. For the AA, ASC and GSH assays non-linear issues were not encountered. Finaly, non-linear ESR response to PM concentrations was solved with backcorrection of ESR signal using

5   linear curve. All together these methodological points allowed avoiding non-additive effects in OP assays and allowed PM samples inter-comparison.

The seasonal variation in OPv (OP measurements expressed per $m^3$) is reported in Figure 2 (A, C, E, G, and I). Overall, higher and more variable OPv values were observed in the cold period than during the warm period where overall values remain low and close to each other. However, patterns did differ depending on the OP measurement method used.

10   Significant differences between cold and warm periods were observed for the DTT, AA, GSH and ASC assays (Figure 2 B, D, F, H and J), but not for OP ESR (Figure 2 F). For the ESR assay, the yearly pattern could be attributed to a possible scavenging effect of $HO^{\bullet}$ by carbonaceous materials (Hellack et al., 2015) leading to the observed non-seasonal variation. Table 1 presents the median ratios calculated between the two periods. OP ESRv values were equivalent in the 2 periods with a ratio of the medians of 0.8 (close to 1 and the interquartile range includes 1) whilet for the other assays, the ratio was

15   always higher than 2. The OPv DTT was only two times higher during the cold period compared with the warm period, whilst higher contrasts of 5, 7.1 and 8.3 were found using the GSH, AA and ASC assays, respectively.

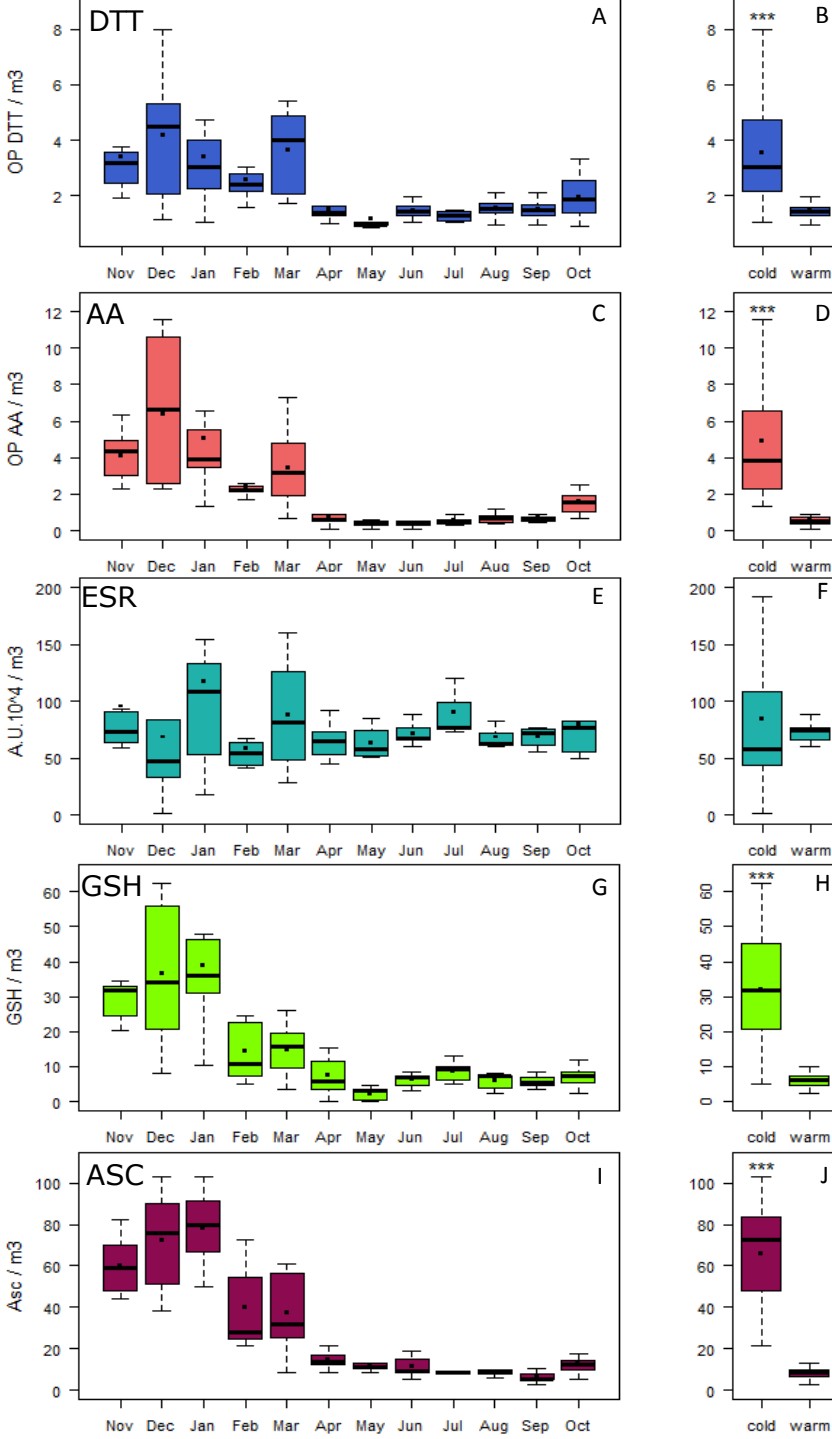

**Figure 2: Seasonal variation of the five OPv (A= OP DTT, C= OP AA, E= ESR, G= OP GSH, I= OP ASC) and OPv differences between cold and warm periods (B= OP DTT, D= OP AA, F= OP ESR, H= OP GSH, J= OP ASC). Boxplot representations with medians (horizontal line) and means (black square). *** p <0.001; nonparametric Mann-Whitney test.**

### 4.3 Comparison of the OP measurement assays

Spearman correlations ($r_s$) between the assays were calculated over the sampling year. Correlations were considered as strong for $r_s > 0.70$ and moderate for $r_s$ ($> 0.45$ and $<0.70$) which are commonly criteria that can be found in the literature (Janssen et al., 2014; Yang et al., 2014). Table 2 shows that OPv values are all strongly correlated ($r_s > 0.7$) with the exception of OP ESRv values for which correlations ranged from 0.16 (OP ASCv) to 0.45 (OP DTTv). All assays are also strongly correlated with $PM_{10}$ ($r_s > 0.77$), except for OP ESRv ($r_s = 0.33$). The strong correlations between OP DTTv and OP AAv are in agreement with another study on $PM_{10}$ (Janssen et al., 2014) whereas they were not observed by Fang et al. (2016) on $PM_{2.5}$. The weak correlations found between the OP ESR assay and both OP DTT and OP GSH assays are also in agreement with other studies on $PM_{2.5}$ (Künzli et al., 2006; Yang et al., 2014). However, the OP ESR assay is usually highly correlated with OP AA and ASC assays in both $PM_{10}$ and $PM_{2.5}$ studies (Janssen et al., 2014; Künzli et al., 2006; Yang et al., 2014).

Table 2: Spearman correlation of the five OPv measurement assays over the sampling year (*** p < 0.001 level, ** p<0.01 level, * p<0.05 level).

|  | $PM_{10}$ n=98 | OP DTTv n=98 | OP AAv n=98 | OP ESRv n=75 | OP GSHv n=98 | OP ASCv n=95 |
|---|---|---|---|---|---|---|
| $PM_{10}$ |  |  |  |  |  |  |
| OP DTTv | 0.88 *** |  |  |  |  |  |
| OP AAv | 0.81 *** | 0.82 *** |  |  |  |  |
| OP ESRv | 0.33 ** | 0.45 *** | 0.27 * |  |  |  |
| OP GSHv | 0.77 *** | 0.73 *** | 0.80 *** | 0.26 * |  |  |
| OP ASCv | 0.80 *** | 0.72 *** | 0.82 *** | 0.16 | 0.80 *** |  |

As shown in Table 3, these correlations also vary with the period, with higher $r_s$ values during cold period ($0.54 < r_s < 0.92$) than in the warm period for which the highest correlation ($r_s = 0.65$) was found between OP DTTv and OP AAv. Temporal variations of correlations were also observed by Fang et al. (2016) on $PM_{2.5}$. During the cold period, $r_s$ between OP methods and $PM_{10}$ ranged from moderate (0.59) for OP ESRv to strong ($> 0.7$) for the other OP measurements. During the warm period, a strong correlation was observed between $PM_{10}$ and OP DTTv.

**Table 3 : Spearman correlation of the OPv in the cold (upper part, blue shaded) and the warm period (*** p < 0.001 level, ** p<0.01 level, * p<0.05 level, [a]n=30 (cold period), n=29 (warm period), [b]n=30 (cold period), n=14 (warm period), [c]n=27 (cold period), n=29 (warm period)).**

| | $PM_{10}$ | OP DTTv [a] | OP AAv [a] | OP ESRv [b] | OP GSHv [a] | OP ASCv [c] |
|---|---|---|---|---|---|---|
| $PM_{10}$ | | 0.92*** | 0.91*** | 0.59*** | 0.87*** | 0.90*** |
| OP DTTv | 0.71*** | | 0.89*** | 0.61*** | 0.79*** | 0.72*** |
| OP AAv | 0.43* | 0.65*** | | 0.54*** | 0.85*** | 0.79*** |
| OP ESRv | 0.088 | 0.17 | 0.36 | | 0.56** | 0.59** |
| OP GSHv | 0.44* | 0.29 | 0.36. | 0.63* | | 0.92*** |
| OP ASCv | 0.38* | 0.37* | -0.072 | -0.29 | 0.17 | |

To better understand the evolution of OP results, OPv were related to the chemical composition of $PM_{10}$ by using both univariate and multiple regressions analyses.

### 4.4 Univariate data analyses

Spearman correlations between OPv and the chemical composition of $PM_{10}$ (expressed in $ng/m^3$ or $\mu g/m^3$) were calculated over the sampling year and are summarized in Table 4. The highest cumulative score of correlations above 0.45 is seen for OP DTTv (37/48; detailed information on cumulative scores is available in the SI, section 4). Conversely, few correlations above 0.45 (5/48) are seen for OP ESRv. The other 4 assays (OP AAv, OP GSHv and OP ASCv) exhibit very similar correlations with the chemical species.

Whereas correlations are evident between OP DTTv, OP AAv, OP GSHv, OP ASCv and some organic and inorganic species, in agreement with other studies regarding both $PM_{2.5}$ and $PM_{10}$ (Janssen et al., 2014; Saffari et al., 2014; Yang et al., 2014), for OP ESRv correlations are limited to transition metals (Cr, Cu, Mo, Zr) and Ba. All these metals may be associated with traffic emissions, including brake wear (Hulskotte et al., 2014; Sanders et al., 2003). These results can be explained by the specificity of the ESR assay towards hydroxyl radical generation and in the case of Ba (not redox-active) linked to its co emission with the former redox active compounds. These results agree with the study of Boogaard et al. (2012) where the OP ESRv of PM collected near major urban road were highly correlated with Cr, Cu and Ba. However, OP ESRv is also associated to organic compounds (OC, PAH) in other studies (Janssen et al., 2014; Yang et al., 2014).

While many chemical species have a cumulative score of 4/5 over the sampling year, Cu is the only species with moderate to strong correlations with all OP measurements (N=5/5). Several studies, including one in a subway system (Moreno et al., 2017) already pointed out the high impact of Cu concentrations on OP.

**Table 4 : Spearman correlations between the OPv measurement assays and the chemical composition of PM10 (dark pink shaded : >0.7, light pink shaded : >0.45) (\*\*\* p < 0.001 level, \*\* p<0.01 level, \* p<0.05 level, . p<0.10 level. N= cumulative score of correlations (moderate and strong)).**

| | DTT | AA | ESR | GSH | ASC | N (/5) |
|---|---|---|---|---|---|---|
| $Cl^-$ | 0.68\*\*\* | 0.73\*\*\* | | 0.61\*\*\* | 0.77\*\*\* | 4 |
| $NO_3^-$ | 0.61\*\*\* | 0.59\*\*\* | | 0.58\*\*\* | 0.74\*\*\* | 4 |
| $SO_4^{2-}$ | 0.33\*\*\* | | 0.27\* | | | |
| $Na^+$ | 0.47\*\*\* | 0.42\*\*\* | | 0.34\*\*\* | 0.46\*\*\* | 2 |
| $NH_4^+$ | 0.38\*\*\* | 0.25\* | | 0.32\*\* | 0.33\*\*\* | |
| $K^+$ | 0.83\*\*\* | 0.85\*\*\* | 0.34\*\* | 0.79\*\*\* | 0.77\*\*\* | 4 |
| $Mg^{2+}$ | 0.56\*\*\* | 0.30\*\* | 0.43\*\*\* | 0.30\*\* | 0.28\*\* | 1 |
| $Ca^{2+}$ | 0.53\*\*\* | 0.24\* | 0.32\*\* | 0.21\* | 0.32\*\* | 1 |
| Al | | | 0.24\* | | | |
| As | 0.62\*\*\* | 0.44\*\*\* | 0.31\*\* | 0.43\*\*\* | 0.41\*\*\* | 1 |
| Ba | 0.68\*\*\* | 0.46\*\*\* | 0.57\*\*\* | 0.42\*\*\* | 0.35\*\*\* | 3 |
| Cd | 0.64\*\*\* | 0.65\*\*\* | | 0.63\*\*\* | 0.72\*\*\* | 4 |
| Ce | 0.50\*\*\* | 0.30\*\* | 0.35\*\* | 0.33\*\*\* | 0.27\*\* | 1 |
| Cr | 0.55\*\*\* | 0.35\*\*\* | 0.54\*\*\* | 0.34\*\*\* | | 2 |
| Cu | 0.87\*\*\* | 0.76\*\*\* | 0.48\*\*\* | 0.70\*\*\* | 0.64\*\*\* | 5 |
| Fe | 0.71\*\*\* | 0.48\*\*\* | 0.44\*\*\* | 0.43\*\*\* | 0.38\*\*\* | 2 |
| La | 0.44\*\*\* | 0.23\* | 0.37\*\* | 0.29\*\* | 0.27\*\* | |
| Li | 0.22\* | | | | | |
| Mn | 0.53\*\*\* | 0.22\* | 0.41\*\*\* | 0.22\* | 0.21\* | 1 |
| Mo | 0.65\*\*\* | 0.38\*\*\* | 0.46\*\*\* | 0.40\*\*\* | 0.27\*\* | 2 |
| Ni | 0.44\*\*\* | | 0.37\*\* | | | |
| Pb | 0.61\*\*\* | 0.42\*\*\* | 0.30\*\* | 0.43\*\*\* | 0.37\*\*\* | 1 |
| Rb | 0.84\*\*\* | 0.76\*\*\* | 0.33\*\* | 0.70\*\*\* | 0.71\*\*\* | 4 |
| Sb | 0.79\*\*\* | 0.66\*\*\* | 0.43\*\*\* | 0.59\*\*\* | 0.52\*\*\* | 4 |
| Sn | 0.70\*\*\* | 0.70\*\*\* | 0.29\* | 0.69\*\*\* | 0.83\*\*\* | 4 |
| Sr | 0.55\*\*\* | 0.29\*\* | 0.44\*\*\* | 0.27\*\* | 0.28\*\* | 1 |
| Ti | 0.36\*\*\* | | 0.40\*\*\* | | | |
| V | | -0.28\*\* | 0.35\*\* | -0.25\* | -0.31\*\* | |
| Zn | 0.84\*\*\* | 0.66\*\*\* | 0.40\*\*\* | 0.65\*\*\* | 0.64\*\*\* | 4 |
| Zr | 0.70\*\*\* | 0.50\*\*\* | 0.50\*\*\* | 0.48\*\*\* | 0.33\*\*\* | 4 |
| BC | 0.82\*\*\* | 0.90\*\*\* | 0.31\* | 0.75\*\*\* | 0.78\*\*\* | 4 |
| $BC_{wb}$ | 0.77\*\*\* | 0.93\*\*\* | 0.27\* | 0.74\*\*\* | 0.87\*\*\* | 4 |
| $BC_{ff}$ | 0.79\*\*\* | 0.84\*\*\* | 0.33\*\* | 0.70\*\*\* | 0.70\*\*\* | 4 |
| OC | 0.83\*\*\* | 0.87\*\*\* | 0.26\* | 0.81\*\*\* | 0.86\*\*\* | 4 |
| DOC | 0.79\*\*\* | 0.87\*\*\* | 0.29\* | 0.86\*\*\* | 0.87\*\*\* | 4 |
| HULIS | 0.85\*\*\* | 0.87\*\*\* | 0.26\* | 0.81\*\*\* | 0.86\*\*\* | 4 |
| EC | 0.82\*\*\* | 0.93\*\*\* | 0.25\* | 0.79\*\*\* | 0.84\*\*\* | 4 |
| TC | 0.84\*\*\* | 0.91\*\*\* | 0.26\* | 0.82\*\*\* | 0.87\*\*\* | 4 |
| MSA | -0.26\*\* | -0.53\*\*\* | | -0.36\*\*\* | -0.34\*\*\* | |
| Oxalate | | | | | | |
| ∑ polyols | | -0.33\*\*\* | 0.38\*\*\* | -0.27\*\* | -0.51\*\*\* | |
| ∑ monosaccharides | 0.74\*\*\* | 0.94\*\*\* | | 0.78\*\*\* | 0.87\*\*\* | 4 |
| ∑ PAHs | 0.77\*\*\* | 0.92\*\*\* | | 0.78\*\*\* | 0.88\*\*\* | 4 |
| ∑ alkanes | 0.63\*\*\* | 0.61\*\*\* | | 0.54\*\*\* | 0.77\*\*\* | 4 |
| ∑ methyl-PAHs | 0.67\*\*\* | 0.85\*\*\* | | 0.72\*\*\* | 0.87\*\*\* | 4 |
| ∑ PASHs | 0.60\*\*\* | 0.76\*\*\* | | 0.71\*\*\* | 0.76\*\*\* | 4 |
| ∑ hopanes | 0.66\*\*\* | 0.67\*\*\* | | 0.58\*\*\* | 0.79\*\*\* | 4 |
| ∑ methoxyphenols | 0.72\*\*\* | 0.93\* | | 0.77\*\*\* | 0.82\*\*\* | 4 |
| **N (/48)** | **37** | **27** | **5** | **25** | **25** | |

## 4.5 Multiple linear regression models

Multiple linear regressions were investigated in order to obtain further insight into the set of chemical species that can be among the dominant factors to the different OP measurements and for OP variations within time.

Such analysis requires the removal of extreme values, and also require transformations of the OP data in order to obtain distributions as close as possible to the Normal one (detailed information about data set preparation can be found in the supporting information Section 5 and Table S2). The multiple linear regression models were obtained using linear model (lm) function in R software. Forward variable selection and BIC number criteria were used to select the predictors that give the better explanation of the variance of the OPv (SI Section 5 model realizations and validations). The models obtained, for the set of samples excluding the extreme values, are presented in the following Eq (2, 3, 4, 5, and 6):

$$\log(\text{OP DTTv}) = -0.440 + 0.101 \times \text{Cu} + 0.0259 \times \textstyle\sum\text{PAH} - 0.0247 \times \text{Mg}^{2+}$$
$$+ 0.273 \times \text{As} + 0.318 \times \text{Mo} + 0.00549 \times \text{MSA} + 0.002 \qquad R^2=0.81 \qquad (2)$$
$$\times \textstyle\sum\text{Polyols} + 0.00135 \times \text{Na}^+$$

$$\log(\text{OP AAv}) = 0.433 + 0.000387 \times \textstyle\sum\text{Monosaccharides} + 0.756 \times \text{Ni} - 0.372$$
$$\times \text{Mo} + 0.000538 \times \text{Fe} \qquad R^2=0.93 \qquad (3)$$

$$\log(\text{OP AAv}) = -1.43 + 0.00163 \times \textstyle\sum\text{Monosaccharides} - 0.00171 \times \text{Ca}^{2+}$$
$$+ 1.69 \times \text{Sb} + 0.117 \times \text{Cu} + 0.00208 \times \text{Na}^+ + 0.00119 \times \text{Fe} \qquad R^2=0.87 \qquad (4)$$
$$- 0.0435 \times \text{Zn}$$

$$\text{OP ESRv} = 339953 + 73037 \times \text{Cu} - 6725 \times \textstyle\sum\text{Alkanes} - 441351 \times \text{Cd}$$
$$+ 20795 \times \text{Ti} \qquad R^2=0.62 \qquad (5)$$

$$\sqrt{\text{OPGSHv}} = 1.21 + 0.00161 \times \textstyle\sum\text{Monosaccharides} + 0.210 \times \text{Cu} - 0.0642$$
$$\times \text{Mg}^{2+} + 0.0268 \times \text{MSA} \qquad R^2=0.48 \qquad (6)$$

$$\log(\text{OP ASCv}) = 1.79 + 0.000659 \times \text{NO3}^- - 0.00733 \times \textstyle\sum\text{Polyols}$$
$$- 0.00292 \times \text{Na}^+ + 0.000247 \times \text{OC}^* + 0.103 \times \text{Cu} \qquad R^2=0.74 \qquad (7)$$

Two distinct models were associated for the AA single compound assay since the distribution is highly bi-modal with two normal distributions, one in a cold period and one in a warmer period (SI Section 5 data set preparation). Strong coefficients of determination were found for the OP DTTv, AAv and ASCv ($R^2$ adj > 0.70) indicating that the variance of these OPv was well explained. For the ESRv and GSHv, $R^2$ were lower, 0.62 and 0.48 respectively (SI Table S4).

Next, these models have then been applied to the general data set (Figure S6), but when strongly overestimated, values were removed again (SI Section 5 application to the overall data set). The models tend to underestimate OP values (negative intercept and slope = 0.84) for the ASC assay. For the other assays, the models tend to overestimate high OP values and to underestimate low OP values (negative intercept and slope between 1.11 and 1.51). However, the coefficients of determination range from 0.73 (OP ASCv) to 0.89 (OP AAv). Altogether, these results indicate that on average, the models

correctly represent the OP of PM participating to the ROS exposure during the overall year.

Finally, the contribution of each predictor during cold and warm periods has been investigated (Figure 3) (SI Section 5 contribution of each predictor). The intercepts, attributed to unknown species, were significantly > 0 in all models. Moreover, for some species, a negative contribution was found (Table S4) that can be explained by an antagonist effect of some atmospheric components on OP : soot particles for Hellack et al. (2015), Gram positive bacteria for Samake et al.

(2017) or metal-organic binding interactions (Verma et al., 2012; Tuet et al., 2016; Wang et al., 2017) or because of the weighting assignation of species in the models. During the cold period, traffic tracers or indicators (like Cu, Fe, Mo, Ti, or $\sum$ PAHs) are important positive factors to explain all OP measurements. Also, and with the exception of the ESR assay, biomass burning tracers ($\sum$ Monosaccharides, including levoglucosan) or indicators such as $\sum$ PAHs or OC* (corresponding to total OC minus the molecular species measured) both strongly related to biomass burning emission (Bonvalot et al., 2016;

Chevrier, 2016), are prominent positive factors for all assays for this cold period. These results agree with the observations on the seasonal evolution of OPv, with much larger values in the cold period when biomass burning emissions are dominant. Additionally, literature results on DTT reactivity towards organic compounds indicate higher impact of biomass burning species when compared to other organic compounds (Verma et al., 2015). During the warm period, traffic tracers and especially copper are important positive factors in OP measurements. In addition, road dust and industrial tracers (Zn, $Ca^{2+}$)

for the AA assay, and primary biogenic indicators ($\sum$polyols, MSA) for the DTT and GSH assays seem prominent factors of these OP measurements.

Several limitations can be attributed to this study. Most important, all of these results have been obtained for a specific location and cannot be generalized as chemical composition of $PM_{10}$ strongly differs from one location to another. $PM_{10}$ chemistry is different from $PM_{2.5}$ and the associations reported here are only valid for $PM_{10}$. Some components that might

mainly reside in the coarse mode are positive factors in the multiple linear regression models (e.g Ti in OP ESRv). They can display a different final health impact, since a fraction of $PM_{10}$ does not penetrate all the way to lung. Also, the results of the ESR assay warrant caution due to our back correction of the ESR signal linked to the non-linear response of the assay. Finally, multiple model result for the GSH assay is to be considered with caution since normal distribution was not reached

in the first step of the analysis. Moreover, multiple non-linear regression models should also be investigated since several studies have shown that oxidative potentials from different PM components are not always additive (Tuet et al., 2016; Tuet et al., 2018; Wang et al., 2017; Xiong et al., 2017; Yu et al., 2018). Finally, these analyses are only relevant for $PM_{10}$ when some health studies are now taking $PM_{2.5}$ into account. Additional studies addressing comparison of OP results associated

5    with $PM_{10}$ and $PM_{2.5}$ are needed (Gali et al., 2017; Styszko et al., 2017).

However, all results point out that biomass burning and vehicular emissions in winter are the main sources correlated with ROS generation. In summer, ROS burden is associated to vehicular emissions, biogenic emissions, road dust, and industrial sources. All these results also suggest to associate assays in order to take into account this wide range of OP determinants. To achieve this, the DTT and the AA assays, or the DTT and the RTLF assays, can be associated to get the best information,

10   which is in agreement with other studies on acellular OP measurements (Janssen et al., 2014; Yang et al., 2014). However, a definitive proposition for the best association of assays will most probably come from a final benchmark against epidemiological study outcomes (Fang et al., 2016; Strak et al., 2017; Weichenthal et al., 2016a).

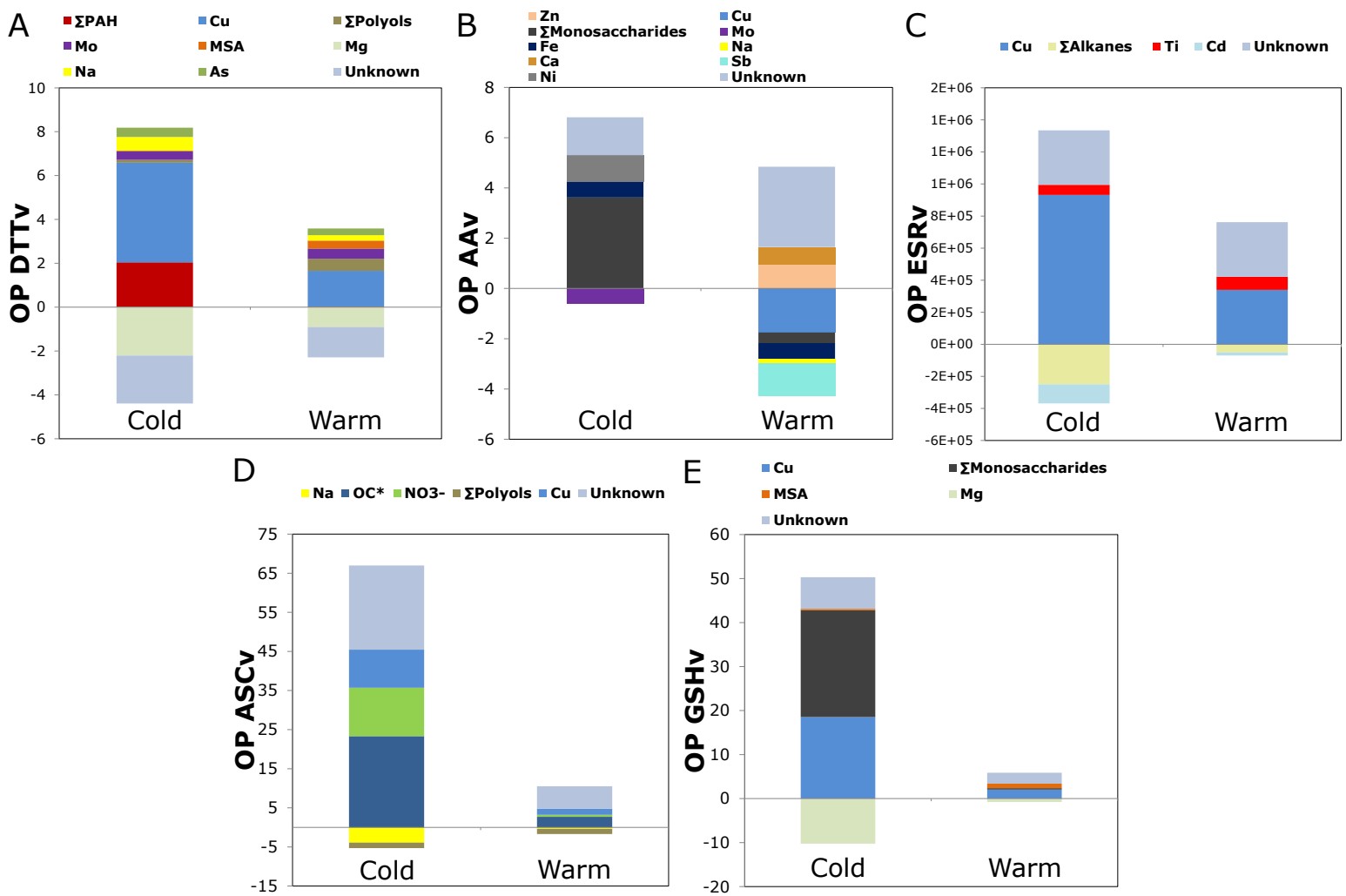

**Figure 3: Multivariable linear regression analyses results: Contribution of predictors during cold and warm periods (A: OP DTTv, B: OP AAv, C: OP ESRv, D: OP GSHv, E: OP ASCv). The intercepts, attributed to unknown species, were significantly > 0 in all models. Some species were assigned with negative contributions, explained by a weighting assignation of species so as to model OP differences between warm and cold periods or a possible antagonist effect of some atmospheric components on OP contribution.**

**5 Conclusion**

Our results show that most of the OP assays were strongly inter-correlated over the sampling year but also these correlations differed when considering specific sampling periods (cold vs warm).

All acellular assays are correlated with a significant number of chemical species when considering nonparametric rank
correlations, especially for the DTT assay. Finally, copper seems to be a unifying determinant in the OP assays.

Evidence is also presented of a seasonal contrast over the sampling period with significantly higher OPv values during winter for the DTT, AA, GSH and ASC assays, which were assigned to biomass burning species by the multiple linear regression models. However, during the cold period and for the DTT assay, $\sum$ PAHs, which can be associated to both traffic and biomass wood burning emissions, were found to be significant factors. The ESRv clearly differs from the other tests as it
did not show seasonal dynamics and presented weaker correlations with other assays and with the chemical species. Nevertheless, ESR assay results are mostly associated with traffic tracer species. Finally, the combination of 2 models was used to fit the results of the AA assay which is necessary to provide the best explanation of OP variance, with a bi-modal distribution of the initial measurements. This indicates that the strong changes in the chemistry of the PM are probably leading to non-linear processes in the link between chemistry and OP.

Overall, these results suggest to combine assays in order to take into account a wide range of determinants of OP. In the case of the Chamonix city, DTT associated with AA assays or DTT combined with RTLF assay were able to provide the most exhaustive information about OP determinants (Cu) and sources associations (biomass burning, vehicular emissions).

Rank correlations and multiple linear regressions are useful tools to determine the most prominent species driving the redox activity of ambient PM. However, to go further in identifying the assay or combination leading to the best information
relying on source dynamics, multiple linear regressions analysis that require large data sets. Finally, more source apportionment approaches through positive matrix factorization methods are needed in order to assign dominant PM emission sources in the OP assays

*Competing interests*. The authors declare no conflict of interest or competing financial interest

*Acknowledgements*. This work was funded in part by Primequal (DECOMBIO program in the Arve Valley, grant ADEME 1362C0028) and by ANSES (ExPOSURE program, grant 2016-CRD-31). The Région Auvergne Rhône-Alpes funded the PhD grant for F. Chevrier. The French research minister funded the PhD grant of A Calas with a Président Award. This
study was also supported by direct funding by IGE (technician salary), the LEFE CHAT (program 863353: "Le PO comme

proxy de l'impact sanitaire"), and the LABEX OSUG@2020 (ANR-10-LABX-56) (both for funding analytical instruments). RENARD (Reseau National de Rpe interDisciplinaire) is kindly acknowledged for access to ESR analysis.

The authors would like to thank Thomas Lacroix, Jean-Charles Francony, Coralie Connès, Vincent Lucaire, and Fanny Masson for their dedicated work for the sample chemical an OP analyses, together with many people from Atmo AURA for collection of samples in the field, and G Brulfert (Atmo AuRA) for strong collaboration on the DECOMBIO Program. G Mocnik and I Jezek took part in the DECOMBIO program with all the measurement by AE33.

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
