# Peer review of "Comparison between five acellular oxidative potential measurement assays performed with detailed chemistry on $PM_{10}$ samples from the city of Chamonix (France)"

_Atmospheric Chemistry and Physics, 2017_

## Referee Comment (RC1) · Anonymous Referee #1 · 2 Jan 2018

This paper reports on comparisons of a number of acellular assays during one year of non-continuous sampling at a single site. This type of study is crucial to help guide selection of assays for future use in health studies and to help interpret existing health studied involving measurement of OP. However, this paper is lacking in many ways. First, why was PM10 used when in most studies reporting on aerosol OP the focus is on PM2.5, as it is in most health studies. There are a large number of citations that are not considered in this work, with some result published work reporting completely opposite to the findings reported here and which the authors seem unaware. This is

a major issue that must be corrected prior to publication, specifically a more complete Introduction and more complete discussion of results in the context of published work. It should also be made clear that when comparisons are being made to other studies, that those studies also were reporting findings based on PM10 (more on this below). Mixing results from PM10 and PM2.5 studies leads to confusion.

Specific comments:

The Introduction is missing many key citations. This includes: Page 2, line 11. There are more acellular assays then listed. What about those measuring OH production, eg, [Charrier and Anastasio, 2011; Vidrio et al., 2009]?

Page 2 lines 18-23: A whole series of papers is not discussed for data collected over various seasons that compares two assays (DTT and AA) [Fang et al., 2016] and compares the assays to chemical species or sources [Verma et al., 2012; Verma et al., 2015a; Verma et al., 2015b; Verma et al., 2014].

Page 2 lines 33, there are more health studies than cited by the author, which are useful to cite in this journal since many readers will not be familiar with the health journals where much of this work is published. Ie, [Abrams et al., 2017; Atkinson et al., 2016; Bates et al., 2015; Strak et al., 2012; Weichenthal et al., 2016a; Weichenthal et al., 2016b]

In my view it is unfortunate that the authors choose to measure and compare PM10 since the composition of PM2.5 vs PM10 is often very different and PM10 less often used in health studies – which is the whole point of this work. It would be helpful to know specifically why PM10 was studied if the goal is to develop insights on assays used in health studies, and how, if any, can these results be applied to PM2.5, which is what the vast majority of OP measurements report. For example, it would be very useful to state what was used (PM10 or PM2.5) in the cited OP-health studies.

When comparing the results of this study involving PM10 to others, which may or may

not be measurements of PM10, or even referring to other studies, it must be clear what is being compared. For example, page 11 line 6 and line 9-10, page 12 line 10 and line 16, and page 16 line 2; the results of this study are claimed to be in agreement with Janssen et al, 2014, Yang et al., 2014, etc, but Janssen measured PM10 and PM2.5; are the results being compared to just the Janssen PM10 results? Yang did only PM2.5, so how can these results be directly compared to Yang without noting this important difference? Much more care must be considered given that this work is only PM10.

Page 2 line 12, It is understood that doing a bulk analysis integrates over size, but the way it is stated (Oxidative potential tests for airborne particles integrate several of biologically relevant properties (e.g. size, and chemical composition) likely to drive PM toxicity.) makes it sound like a positive attribute of the assay, it is not. The advantage of these assays is they integrate chemical species in the particles that may contribute to OP, a unifying property potentially linked to the particles toxicity through oxidative stress. Integrating over size is likely not advantageous as in the case of PM10 it mixes very different aerosol sources (chemical components).

How exactly were the PM10 filter samples collected, eg, what type of filter sampling system at what flow rate.

Page 3 to 6. It would be valuable to know if the assays are performed in a manor exactly consistent with given protocols for each assay. This is given to some extent, but more explicit statements on this would clarify things. For example, does the DTT assay follow the protocol of Cho et al, [Cho et al., 2005], etc?

Page 9 line 2, what does more dispersed data mean? Higher standard deviation?

Page 11, line 6, It states, that : OP DTTv and OP AAv are in agreement with another study (Janssen et al., 2014). However this was not found in another study, [Fang et al., 2016].

Univariate analysis of OP with metals. (page 12). Given that total elemental metal concentrations were measured, not water-soluble or speciated metal ions, how can one link these metals to OP assays through redox activity since: 1) Total (elemental) metal concentration is not necessarily correlated with the soluble metal concentration. 2) Only the soluble metal is involved in the redox reactions. The total metals can only be used to identify sources (assuming source profiles were based on total metals).

Page 15, line 10, what does "PM participating to the background"... mean

Page 15. Line 14, some other studies show antagonistic effects on OP that one may wish to consider, see [Wang et al., 2017; Xiong et al., 2017].

Page 15 line 28: What exactly does this line mean? "Additionally, PM extractions were realized only for the DTT and the AA assays which can lead to a difficulty in the results comparison."

Page 16, Line 2, this conclusion is opposite of Fang et al [Fang et al., 2016], which should be noted and possibly discuss possible reasons why.

Page 15 and 16, I would say the biggest limitation is the use of PM10 for this study instead of PM2.5. This would be a good place in the paper to discuss this (see comments above).

Fang et al, [Fang et al., 2016] did exactly what the last line of the main text states, and it is never discussed, although the paper is cited.

Page 18, line 21 and 22, a source apportionment like that which has already been done, but never cited? See [Bates et al., 2015; Fang et al., 2017; Verma et al., 2014]

References:

Abrams, J., R. J. Weber, M. Klein, S. E. Samat, H. H. Chang, M. J. Strickland, V. Verma, T. Fang, J. T. Bates, J. A. Mulholland, A. G. Russell, and P. E. Tolbert (2017), Associations between ambient fine particulate oxidative potential and cardiorespiratory emergency department visits, Envir. Health Perspectives, https://doi.org/10.1289/EHP1545.

Atkinson, R. W., E. Samoli, A. Analitis, G. W. Fuller, D. C. Green, H. R. Anderson, E. Purdie, C. Dunster, L. Aitlhadj, F. J. Kelly, and I. S. Mudway (2016), Short-term associations between particle oxidative potential and daily mortality and hospital admissions in London, Intern. J. of Hygiene and Envir. Health, 219, 566-572.

Bates, J. T., R. J. Weber, J. Abrams, V. Verma, T. Fang, M. Klein, M. J. Strckland, S. Sarnat, H. Chang, J. A. Mulholland, P. E. Tolbert, and A. G. Russell (2015), Reactive Oxygen Species in Atmospheric Particulate Matter Suggest a Link to Cardiorespiratory Effects, Envir. Sci. Technol, 49, 13605-13612.

Charrier, J. G., and C. Anastasio (2011), Impacts of antioxidants on hydroxyl radical production from individual and mixed transition metals in a surrogate lung fluid, Atmos. Env., 24, 7555-7562.

Cho, A. K., C. Sioutas, A. H. Miguel, Y. Kumagai, D. A. Schmitz, M. Singh, A. Eiguren-Fernandez, and J. R. Froines (2005), Redox activity of airborne particulate matter at different sites in the Los Angeles Basin, Environ. Res., 99, 40-47.

Fang, T., L. Zeng, D. Gao, V. Verma, A. Stefaniak, and R. J. Weber (2017), Ambient Size Distributions and Lung Deposition of Aerosol Oxidative Potential: A Contrast Between Soluble and Insoluble Particles Envir. Sci. Technol, DOI: 10.1021/acs.est.7b01536.

Fang, T., V. Verma, J. T. Bates, J. Abrams, M. Klein, M. J. Strickland, S. E. Sarnat, H. H. Chang, J. A. Mulholland, P. E. Tolbert, A. G. Russell, and R. J. Weber (2016), Oxidative potential of ambient water-soluble PM2.5 in the southeastern United States: contrasts in sources and health associations between ascorbic acid (AA) and dithiothreitol (DTT) assays, Atmos. Chem. Phys., 16, 3865-3879.

Strak, M. J., N. A. H. Janssen, K. J. Godri, I. Gosens, I. S. Mudway, F. R. Cassee, E. Lebret, F. J. Kelly, R. M. Harrison, B. Brunekreef, M. Steenhof, and G. Hoek (2012), Respiratory Health Effects of Airborne Particulate Matter: The Role of Particle Size,

Composition, and Oxidative Potential — The RAPTES Project, Envir. Health Perspectives, 102(8), 1183-1189.

Verma, V., R. Rico-Martinez, N. Kotra, L. King, J. Liu, T. W. Snell, and R. J. Weber (2012), Contribution of water-soluble and insoluble components and their hydrophobic/hydrophilic sub-fractions on the ROS-generating potential of fine ambient aerosols, Environ. Sci. Technol., 46, 11384-11392.

Verma, V., Y. Wang, R. El-Afifi, T. Fang, J. Rowland, A. G. Russell, and R. J. Weber (2015a), Fractionating Ambient Humic-like Substances (HULIS) for their reactive oxygen species activity - Assessing the importance of quinones and atmospheric aging on particulate matter toxicity, Atmos. Env., 120, 351-359.

Verma, V., T. Fang, X. Lu, R. E. Peltier, A. Russell, N. Lee, and R. J. Weber (2015b), Organic Aerosols Associated with the Generation of Reactive Oxygen Species (ROS) by Water-Soluble PM2.5, Environ. Sci. Technol., 49(7), 4646-4656.

Verma, V., T. Fang, H. Guo, L. King, J. T. Bates, R. E. Peltier, E. Edgerton, A. J. Russell, and R. J. Weber (2014), Reactive Oxygen Species Associated with Water-Soluble PM2.5 in the Southeastern United States: Spatiotemporal Trends and Source Apportionment, Atm. Chem. Phys., 14, 12915-12930.

Vidrio, E., C. H. Phual, A. M. Dillner, and C. Anastasio (2009), Generation of Hydroxyl Radicals from Ambient Fine Particles in a Surrogate Lung Fluid Solution, Environ. Sci. Technol., 43, 922-927.

Wang, S., J. Ye, R. Soong, B. Wu, L. Yu, A. Simpson, and A. W. H. Chan (2017), Relationship between Chemical Composition and Oxidative Potential of Secondary Organic Aerosol from Polycyclic Aromatic Hydrocarbons, Atm. Chem. Phys., https://doi.org/10.5194/acp-2017-1012.

Weichenthal, S. A., E. Lavigne, G. J. Evans, K. J. G. Pollitt, and R. T. Burnett (2016a), PM2.5 and Emergency Room Visits for Respiratory Illness: Effect Modification by Oxidative Potential, Am J Resp Crit Care Med, 194, 577-586.

Weichenthal, S. A., D. L. Crouse, L. Pinault, K. Godri-Pollitt, W. Bavigne, G. Evans, A. v. Donkelaar, R. V. Martin, and R. T. Burnett (2016b), Oxidative burden of fine particulate air pollution and risk of cause-specific mortality in the Canadian Census Health and Environment Cohort (CanCHEC), Envir. Res., 146, 92-99.

Xiong, Q., H. Yu, R. Wang, J. Wei, and V. Verma (2017), Rethinking The Dithiothre-itol (DTT) Based PM Oxidative Potential: Measuring DTT Consumption versus ROS Generation, Envir. Sci. Technol, 51, 6507-6514.
* * *

---

## Author Comment (AC1) · 30 Jan 2018

1) This paper reports on comparisons of a number of acellular assays during one year of non-continuous sampling at a single site. This type of study is crucial to help guide selection of assays for future use in health studies and to help interpret existing health studied involving measurement of OP. However, this paper is lacking in many ways. First, why was PM10 used when in most studies reporting on aerosol OP the focus is

on PM2.5, as it is in most health studies. There are a large number of citations that are not considered in this work, with some result published work reporting completely opposite to the fidings reported here and which the authors seem unaware. This is a major issue that must be corrected prior to publication, specifically a more complete Introduction and more complete discussion of results in the context of published work. It should also be made clear that when comparisons are being made to other studies, that those studies also were reporting findings based on PM10 (more on this below). Mixing results from PM10 and PM2.5 studies leads to confusion.

Authors thank this referee for their useful comments. In EU regulation (including France), particulate matter sanitary alerts are based on PM10 measurements. PM10 studied in this paper, originate from the AASQA national French network ( http://www.atmo-france.org/fr/). Some references about this issue have been added in the discussion. In addition, we would like to kindly remind this referee that health impacts from PM are very wide and do not stem only from PM2.5. PM10 contribute also to health outcomes and a large number of recent epidemiological studies use PM10 to predict health associations from particulate pollution, especially in Europe(Griffiths et al, 2016, Piersanti et al, 2016, James, et al. 2015, Segersson et al., 2017, Al-Hemoud,et al , 2018 , the European ongoing ESCAPE study : Cesaroni et al., 2014 etc etc)

Griffiths, C. J., Mudway, I., Wood, H., Marlin, N., Dundas, I., Walton, R., ... & Kelly, F. (2016). P180 Impact of the london low emission zone on children's respiratory health: a sequential yearly cross sectional study 2008–2014. Piersanti, Antonio, Carla Ancona, Giovanna Berti, Ennio Cadum, Luisella Ciancarella, Ilaria D'Elia, Francesco Forastiere, and Gaia Righini. "Health impact of air pollution on Italy: main findings of VIIAS and MED HISS projects." (2016). James, K., Forsen, A., Strand, M., & Cicutto, L. (2015). PM10 Concentrations And Asthma Related Health Services Use In A Rural Community In Colorado. In C15. NOVEL EPIDEMIOLOGY OF ASTHMA AND COPD (pp. A3902-A3902). American Thoracic Society. Segersson, David, Kristina Eneroth, Lars Gidhagen, Christer Johansson, Gunnar Omstedt, Anders Engström Nylén, and Bertil Forsberg. "Health Impact of PM10, PM2. 5 and Black Carbon Exposure Due to Different Source Sectors in Stockholm, Gothenburg and Umea, Sweden." International journal of environmental research and public health 14, no. 7 (2017): 742. Al-Hemoud, Ali, Ali Al-Dousari, Ahmad Al-Shatti, Ahmed Al-Khayat, Weam Behbehani, and Mariam Malak. "Health Impact Assessment Associated with Exposure to PM10 and Dust Storms in Kuwait." Atmosphere 9, no. 1 (2018): 6. Cesaroni G, Forastiere F, Stafoggia M, Andersen ZJ, Badaloni C, Beelen R, Caracciolo B, de Faire U, Erbel R, Eriksen KT, Fratiglioni L, Galassi C, Hampel R, Heier M, Hennig F, Hilding A, Hoffmann B, Houthuijs D, Jockel KH, Korek M, Lanki T, Leander K, Magnusson PK, Migliore E, Ostenson CG, Overvad K, Pedersen NL, J JP, Penell J, Pershagen G, Pyko A, Raaschou-Nielsen O, Ranzi A, Ricceri F, Sacerdote C, Salomaa V, Swart W, Turunen AW, Vineis P, Weinmayr G, Wolf K, de Hoogh K, Hoek G, Brunekreef B, Peters A (2014) Long term exposure to ambient air pollution and incidence of acute coronary events: prospective cohort study and meta-analysis in 11 European cohorts from the ESCAPE Project. BMJ 348:f7412. doi: 10.1136/bmj.f7412

Specific comments:

2) The Introduction is missing many key citations. This includes: Page 2, line 11. There are more acellular assays then listed. What about those measuring OH production, eg, [Charrier and Anastasio, 2011; Vidrio et al., 2009]?

We agree with this comment of the reviewer, and we modified our sentence as following:

Page 2 line 7: On this basis, probes have been developed over the last decade to quantify the OP of PM as a more refined exposure metric of PM toxicity than PM mass alone (Ayres et al., 2008; Borm et al., 2007). These probes include several acellular assays. The most common consisting in mimicking the consumption of antioxidants (e.g. ascorbic acid (AA), reduced glutathione (GSH)) or surrogates (e.g. dithiothreitol (DTT)), the use of the synthetic human respiratory tract lining fluid (RTLF) system (again to assess antioxidant depletion), probes measuring HO• production or the application of electron spin resonance (ESR) to quantify the ability of PM to induce specific ROS (e.g. HO• radicals).

3) Page 2 lines 18-23: A whole series of papers is not discussed for data collected over various seasons that compares two assays (DTT and AA) [Fang et al., 2016] and compares the assays to chemical species or sources [Verma et al., 2012; Verma et al., 2015a; Verma et al., 2015b; Verma et al., 2014].

We agree with this remark and added references

Page 2 line 18 : Only a small number of studies have compared different acellular OP measurements for a given set of ambient PM samples (Fang et al., 2016; Janssen et al., 2014; Künzli et al., 2006; Szigeti et al., 2015; Visentin et al., 2016; Yang et al., 2014). Yet, fewer studies have compared different assays over a year long period to gain a better understanding of seasonal variability (Fang et al., 2016; Jedynska et al., 2017; Saffari et al., 2014; Szigeti et al., 2015; Yang et al., 2015). Finally, there is little research relating the oxidative capacity of particulate pollution with detailed chemical characterization of ambient PM, in an attempt to identify the PM components or sources that may contribute most to underlying toxicity (Fang et al., 2016; Saffari et al., 2014; Verma et al., 2014; Weber et al., 2018, Kelly et al, 2011).

Verma et al., 2012; Verma et al., 2015a; Verma et al., 2015b were not added for not meeting the requirements of this paragraph "detailed chemical characterization AND long time series. (Verma 2012, is only providing two weeks of analysis, Verma 2015a is only focused on Quinones and Hulis, Verma 2015b is only on organic aerosols).

4) Page 2 lines 33, there are more health studies than cited by the author, which are useful to cite in this journal since many readers will not be familiar with the health journals where much of this work is published. Ie, [Abrams et al., 2017; Atkinson et al., 2016; Bates et al., 2015; Strak et al., 2012; Weichenthal et al., 2016a; Weichenthal et

al., 2016b].

We agree with this remark and added studies that shown positive correlation between OP assays and health outcomes:

Page 2 line 33: All of these assays have shown some correlations with health outcomes in epidemiological studies (Abrams et al., 2017; Bates et al., 2015; Fang et al., 2016; Strak et al., 2017; Weichenthal et al., 2016a, 2016b; Yang et al., 2016).

5) In my view it is unfortunate that the authors choose to measure and compare PM10 since the composition of PM2.5 vs PM10 is often very different and PM10 less often used in health studies – which is the whole point of this work. It would be helpful to know specifically why PM10 was studied if the goal is to develop insights on assays used in health studies, and how, if any, can these results be applied to PM2.5, which is what the vast majority of OP measurements report. For example, it would be very useful to state what was used (PM10 or PM2.5) in the cited OP-health studies.

In Europe, regulations and particulate matter sanitary alert are based on PM10 measurements. PM10 studied in this paper, were collected on filters by the certified associations for the monitoring of the air quality (AASQUA). To fill this gap between studies on PM10 and PM2.5, we need more studies comparing size-segregation effect on OP results (Styszko et al., 2017, Gali et al., 2017).

To clarify, we added the following sentences:

Page 3 line 11: In Europe, particulate matter sanitary alert are based on PM10 measurements.

Page 15 line 26: Several limitations can be attributed to this study. Most important, all of these results have been obtained for a specific location and cannot be generalized as chemical composition of PM10 strongly differs from one location to another. PM10 chemistry is different from PM2.5 and the associations reported here are only valid for PM10. Also, the results of the ESR assay warrant caution due to our back correction of

the ESR signal linked to the non linear response of the assay. Finally, multiple model result for the GSH assay is to be considered with caution since normal distribution was not reached in the first step of the analysis. Finally, these analyses are only relevant for PM10 when some health studies are now taking PM2.5 into account. Additional studies addressing comparison of OP results associated with PM10 and PM2.5 are needed (Styszko et al., 2017, Gali et al., 2017)

Styszko, K., Samek, L., Szramowiat, K., Korzeniewska, A., Kubisty, K., Rakoczy-Lelek, R., ... & Giebl, A. K. (2017). Oxidative potential of PM10 and PM2. 5 collected at high air pollution site related to chemical composition: Krakow case study. Air Quality, Atmosphere & Health, 10(9), 1123-1137. Gali, N. K., Jiang, S. Y., Yang, F., Sun, L., & Ning, Z. (2017). Redox characteristics of size-segregated PM from different public transport microenvironments in Hong Kong. Air Quality, Atmosphere & Health, 10(7), 833-844.

6) When comparing the results of this study involving PM10 to others, which may or may not be measurements of PM10, or even referring to other studies, it must be clear what is being compared. For example, page 11 line 6 and line 9-10, page 12 line 10 and line 16, and page 16 line 2; the results of this study are claimed to be in agreement with Janssen et al, 2014, Yang et al., 2014, etc, but Janssen measured PM10 and PM2.5; are the results being compared to just the Janssen PM10 results? Yang did only PM2.5, so how can these results be directly compared to Yang without noting this important difference?

We added some clarification following this remark:

Page 11 line 5 : The strong correlations between OP DTTv and OP AAv are in agreement with another study on PM10 (Janssen et al., 2014) whereas they were not observed by Fang et al. (2016) for PM2.5. The weak correlations found between the OP ESR assay and both OP DTT and OP GSH assays are also in agreement with other studies on PM2.5 (Künzli et al., 2006; Yang et al., 2014). However, the OP ESR assay is usually highly correlated with OP AA and ASC assays in both PM10 and PM2.5 studies (Janssen et al., 2014; Künzli et al., 2006; Yang et al., 2014).

Page 12 line 12 : Whereas correlations are evident between OP DTTv, OP AAv, OP GSHv, OP ASCv and some organic and inorganic species, in agreement with other studies regarding both PM2.5 and PM10 (Janssen et al., 2014; Saffari et al., 2014; Yang et al., 2014),   7) Much more care must be considered given that this work is only PM10.

We added this remark in the limitations of the study

Page 15 line 26: Several limitations can be attributed to this study. Most important, all of these results have been obtained for a specific location and cannot be generalized as chemical composition of PM10 strongly differs from one location to another. PM10 chemistry is different from PM2.5 and the associations reported here are only valid for PM10. Also, the results of the ESR assay warrant caution due to our back correction of the ESR signal linked to the non linear response of the assay. Finally, multiple model result for the GSH assay is to be considered with caution since normal distribution was not reached in the first step of the analysis. Finally, these analyses are only relevant for PM10 when some health studies are now taking PM2.5 into account. Additional studies addressing comparison of OP results associated with PM10 and PM2.5 are needed (Styszko et al., 2017, Gali et al., 2017)

8) Page 2 line 12, It is understood that doing a bulk analysis integrates over size, but the way it is stated (Oxidative potential tests for airborne particles integrate several of biologically relevant properties (e.g. size, and chemical composition) likely to drive PM toxicity.) makes it sound like a positive attribute of the assay, it is not. The advantage of these assays is they integrate chemical species in the particles that may contribute to OP, a unifying property potentially linked to the particles toxicity through oxidative stress. Integrating over size is likely not advantageous as in the case of PM10 it mixes very different aerosol sources (chemical components).

We apologize for this misunderstanding. Through the following sentence Âń Oxidative potential tests for airborne particles integrate several of biologically relevant properties (e.g. size, and chemical composition) likely to drive PM toxicity.", we wanted to refer to this previous statement from Ayres " oxidative potential may integrate various PM characteristics (size, surface area and composition) into a single biologically relevant measure of toxicity," (Ayres, 2008).

J.G. Ayres, P. Borm, F.R. Cassee, V. Castranova, K. Donaldson, A. Ghio, R.M. Harrison, R. Hider, F. Kelly, I.M. Kooter, F. Marano, R.L. Maynard, I. Mudway, A. Nel, C. Sioutas, S. Smith, A. Baeza-Squiban, A. Cho, S. Duggan, J. Froines Evaluating the toxicity of airborne particulate matter and nanoparticles by measuring oxidative stress potential—a workshop report and consensus statement. Inhalation Toxicology, 20 (2008), pp. 75-99

We rephrased this sentence as following/

Page 2 line 12 : Oxidative potential can be considered as an integrative metric of PM characteristics (size, composition, surface area...) potentially linked to the particles toxicity through oxidative stress "

9) How exactly were the PM10 filter samples collected, eg, what type of filter sampling system at what flow rate.

All the information about PM10 sampling is indicated in the material and methods section:

Page 3 line 18: Briefly, ambient particles were collected by filtration during 24 h (24 $\times$ 30 m3.h-1) with a DIGITEL DA-80 on 150 mm quartz filters (Tissuquartz Pallflex).using the European standard protocol NF EN 16450.

10) Page 3 to 6. It would be valuable to know if the assays are performed in a manor exactly consistent with given protocols for each assay. This is given to some extent, but more explicit statements on this would clarify things. For example, does the DTT

assay follow the protocol of Cho et al, [Cho et al., 2005], etc?

The following sentences have been modified:

Page 4 line 19: A semi-automated procedure was used with a plate-reader TECAN spectrophotometer Infinite® M 200 pro and 96 well CELLSTAR® multiwall plates from Greiner bio-one®, the assay was modified from the DTT assay of Cho et al. (2005). Page 5 line 13: As for the DTT assay, a semi-automated procedure using the same plate reader was applied using Greiner UV-Star® 96 well plates, this assay was based on the modified assay from Zielinski et al. (1999) and Mudway et al. (2004).

11) Page 9 line 2, what does more dispersed data mean? Higher standard deviation?

We modified the sentence as following:

Page 9 line 7 : Overall, higher and more variable OPv values were observed in the cold period than during the warm period where overall values remain low and close to each other.

12) Page 11, line 6, It states, that : OP DTTv and OP AAv are in agreement with another study (Janssen et al., 2014). However this was not found in another study, [Fang et al.,2016].

Page 11 line 5: The strong correlations between OP DTTv and OP AAv are in agreement with another study on PM10 (Janssen et al., 2014) whereas they were not observed by Fang et al. (2016) for PM2.5.

We have now discussed more of the study by Fang et al. 2016:

Page 11 line: 15/ As shown in Table 3, these correlations also vary with the period, with higher rs values during cold period (0.54 < rs < 0.92) than in the warm period for which the highest correlation (rs = 0.65) was found between OP DTTv and OP AAv. Temporal variations of correlations were also observed by Fang et al. (2016) on PM2.5. During the cold period, rs between OP methods and PM10 ranged from moderate (0.59) for

OP ESRv to strong (> 0.7) for the other OP measurements. During the warm period, a strong correlation was observed between PM10 and OP DTTv.

13) Univariate analysis of OP with metals. (page 12). Given that total elemental metal concentrations were measured, not water-soluble or speciated metal ions, how can one link these metals to OP assays through redox activity since: 1) Total (elemental) metal concentration is not necessarily correlated with the soluble metal concentration. 2) Only the soluble metal is involved in the redox reactions. The total metals can only be used to identify sources (assuming source profiles were based on total metals).

It's true that total elemental metal concentration is not necessarily correlated with the soluble metal concentration. However, , insoluble metals may also contribute to overall oxidant stress. Insoluble nanoparticles (e.g CuO) complexes and some insoluble metals may lead to positive OP without being soluble by other mechanisms (Calas et al., 2017, Huang et al, 2016, Uzu et al., 2011). Verma et al 2012, showed also that insoluble organics species may contribute to OP. Finally, even if they are questionable and partially answer to the issue of elicits OP drivers, we assume that univariate analysis are useful as a first step to target sources.

Huang, W., Zhang, Y., Zhang, Y., Zeng, L., Dong, H., Huo, P., ... & Schauer, J. J. (2016). Development of an automated sampling-analysis system for simultaneous measurement of reactive oxygen species (ROS) in gas and particle phases: GAC-ROS. Atmospheric Environment, 134, 18-26. Calas, A., Uzu, G., Martins, J. M. F., Voisin, D., Spadini, L., Lacroix, T. and Jaffrezo, J.: The importance of simulated lung fluid ( SLF ) extractions for a more relevant evaluation of the oxidative potential of particulate matter, Sci. Reports, (August), 1–12, doi:10.1038/s41598-017-11979-3, 2017. Uzu, G., Sauvain, J. J., Baeza-Squiban, A., Riediker, M., Sanchez, M., Hohl, S., Val, S., Tack, K., Denys, S., Pradère, P. and Dumat, C.: In vitro assessment of the pulmonary toxicity and gastric availability of lead-rich particles from a lead recycling plant, Environ. Sci. Technol., 45(18), 7888–7895, doi:10.1021/es200374c, 2011. Verma, V., R. Rico-Martinez, N. Kotra, L. King, J. Liu, T. W. Snell, and R. J. Weber (2012), Contribution of water-soluble and insoluble components and their hydrophobic/hydrophilic sub-fractions on the ROS-generating potential of iňĄne ambient aerosols,Environ. Sci. Technol., 46, 11384-11392.

14) Page 15, line 10, what does "PM participating to the background". . . mean

We modified our sentence as follow: Page 15 line 9: Altogether, these results indicate that on average, the models correctly represent the OP of PM participating to the ROS exposure during the overall year.

15) Page 15. Line 14, some other studies show antagonistic effects on OP that one may wish to consider, see [Wang et al., 2017; Xiong et al., 2017].

Thank you for this comment, the study of Wang et al was added as follow:

Page 15 line 12: The intercepts, attributed to unknown species, were significantly > 0 in all models. Moreover, for some species, a negative contribution was found (Table S4) that can be explained by an antagonist effect of some atmospheric components on OP (Hellack et al., 2015; Samake et al., 2017; Wang et al., 2017) or because of the weighting assignation of species in the models.

16) Page 15 line 28: What exactly does this line mean? "Additionally, PM extractions were realized only for the DTT and the AA assays which can lead to a difiňĄculty in the results comparison." . This sentence was removed.

17) Page 16, Line 2, this conclusion is opposite of Fang et al [Fang et al., 2016], which should be noted and possibly discuss possible reasons why.

Page 16 line 4: To achieve this, the DTT and the AA assays, or the DTT and the RTLF assays, can be associated to get the best information, which is in agreement with other studies on acellular OP measurements (Janssen et al., 2014; Yang et al., 2014). However, a definitive proposition for the best association of assays will most probably come from a final benchmark against epidemiological study outcomes (Fang et al., 2016; Strak et al., 2017; Weichenthal et al., 2016a).

18) Page 15 and 16, I would say the biggest limitation is the use of PM10 for this study instead of PM2.5. This would be a good place in the paper to discuss this (see comments above).

See response to point 7.

Page 15 line 26: Several limitations can be attributed to this study. Most important, all of these results have been obtained for a specific location and cannot be generalized as chemical composition of PM10 strongly differs from one location to another. PM10 chemistry is different from PM2.5 and the associations reported here are only valid for PM10. Also, the results of the ESR assay warrant caution due to our back correction of the ESR signal linked to the non linear response of the assay. Finally, multiple model result for the GSH assay is to be considered with caution since normal distribution was not reached in the first step of the analysis. Finally, these analyses are only relevant for PM10 when some health studies are now taking PM2.5 into account. Additional studies addressing comparison of OP results associated with PM10 and PM2.5 are needed (Styszko et al., 2017, Gali et al., 2017)

19) Fang et al, [Fang et al., 2016] did exactly what the last line of the main text states, and it is never discussed, although the paper is cited.

Page 16 line 15 : However, a definitive proposition for the best association of assays will most probably come from a final benchmark against epidemiological study outcomes (Fang et al., 2016; Strak et al., 2017; Weichenthal et al., 2016a).

20) Page 18, line 21 and 22, a source apportionment like that which has already been done, but never cited? See [Bates et al., 2015; Fang et al., 2017; Verma et al., 2014]

Page 18 line 20: Finally, more source apportionment approaches through positive matrix factorization methods are needed in order to assign dominant PM emission sources in the OP assays.

[Figure]

Please also note the supplement to this comment:
https://www.atmos-chem-phys-discuss.net/acp-2017-1062/acp-2017-1062-AC1-
supplement.pdf

**Supplement:**

[revised manuscript text omitted]

---

## Referee Comment (RC2) · Anonymous Referee #2 · 23 Feb 2018

General comments This work presents a comprehensive comparison study of five acellular oxidative potential (OP) assays and examination of correlations of OPs with an extensive list of chemical components in PM10 samples collected over a year-long period in downtown Chamonix, France (sample size n= 98). The work was carefully executed. Of special note is that extractions containing the same final concentration of PM10 mas (i.e., 10 ug/mL) were used for the DTT and AA assays, avoiding the complication caused by non-linear response as to PM concentrations. The paper is well-written and the figures are nicely constructed. This work provides a very nice case study of how

[Figure]

OPs by various assays are associated with different PM10 components. I have a few minor comments listed below.

Specific comments

1. Please describe the sample collection schedule during the one-year period. Were the samples collected following a regular schedule?

2. As the ESR assay only used 75 samples out of the total 98 samples, please include another column in Table S1 to indicate the number of samples in each month used for the ESR assay.

3. Page 15, line13: please list the species that show an antagonist effect. This information is worth a special mention.

4. The samples used in this work were PM10 samples. The coarse PM (PM2.5-10), likely accounting for a significant fraction of PM10, does not penetrate all the way to lung. Some components, such as Ti (likely of dust origin), might mainly reside in the coarse mode. Ti is found to be a positive indicator in the multiple linear regression model equation (Eq. (5) for OP ESRv. There might be a disconnection between OP responses obtained under physiological conditions simulating lung fluid and actual OP impacts from breathing in of PM10. It will be good that the authors comment on this disconnection.

Minor comments

1. It appears both ASC and AA are used as abbreviation to refer to ascorbic acid. Why two abbreviations?

2. The reference "Chevrier 2016" is given in French. Please provide an English translation and also how this reference can be accessed.

3. Please define "DPCC". The first appearance is line 6 on page 4.

4. Page 5, line 9: Is "the DDT assay" supposed to be "the AA assay" instead?

5. Figure S3: is the y-axis label supposed to be "nmol AA/min"?

6. Page 11, lines 2-3: please cite a reference for the criterion for determining whether a correlation is strong or moderate.

7. Table S4: one entry of Mg $\rightarrow$ Mg2+; NO3$\rightarrow$ NO3-; NH4$\rightarrow$NH4+
* * *

---

## Author Comment (AC2) · 22 Mar 2018

Discussion paper Atmos. Chem. Phys. Discuss., https://doi.org/10.5194/acp-2017-1062-RC1, 2018
General comments This work presents a comprehensive comparison study of five acellular oxidative potential (OP) assays and examination of correlations of OPs with an extensive list of chemical components in PM10 samples collected over a year-long period in downtown Chamonix, France (sample size n= 98). The work was carefully executed. Of special note is that extractions containing the same final concentration of PM10 mas (i.e., 10 ug/mL) were used for the DTT and AA assays, avoiding the complication caused by non-linear response as to PM concentrations. The paper is well-written and the figures are nicely constructed. This work provides a very nice case study of how OPs by various assays are associated with different PM10 components. I have a few minor comments listed below.

We really thank you for this positive comment. Some modifications have been made after the review of the first referee; you'll find them in blue on the new main text version.

Specific comments

1) Please describe the sample collection schedule during the one-year period. Were the samples collected following a regular schedule?

Page 3 line 18: Briefly, ambient particles were collected by filtration during 24 h (24 × 30 m3.h-1) with a DIGITEL DA-80 on 150 mm quartz filters (Tissuquartz Pallflex) using the European standard protocol NF EN 16450. DIGITEL DA-80 was automatically program to stock before and after sampled filters, and the samples were then collected every week.

2) As the ESR assay only used 75 samples out of the total 98 samples, please include another column in Table S1 to indicate the number of samples in each month used for the ESR assay.

The table S1 has been modified in order to present number of samples analyzed per assays ( as we could'nt add the table in the plain text of the answer, it was added at the
end of the revieuw as Fig 1)

3) Page 15, line13: please list the species that show an antagonist effect. This information is worth a special mention.

We modified the sentences as follow:

Page 15 line 12: The intercepts, attributed to unknown species, were significantly > 0 in all models. Moreover, for some species, a negative contribution was found (Table S4) that can be explained by an antagonist effect of some atmospheric components on OP : soot particles for Hellack et al. (2015), Gram positive bacteria for Samake et al. (2017) or metal-organic binding interactions for Wang et al. (2017) or because of the weighting assignation of species in the models.

4) The samples used in this work were PM10 samples. The coarse PM (PM2.5-10), likely accounting for a significant fraction of PM10, does not penetrate all the way to lung. Some components, such as Ti (likely of dust origin), might mainly reside in the coarse mode. Ti is found to be a positive indicator in the multiple linear regression model equation (Eq. (5) for OP ESRv. There might be a disconnection between OP responses obtained under physiological conditions simulating lung fluid and actual OP impacts from breathing in of PM10. It will be good that the authors comment on this disconnection.

We added limitations of our study about the distinction of PM10 and PM2.5:

Page 15 line 26: Several limitations can be attributed to this study. Most important, all of these results have been obtained for a specific location and cannot be generalized as chemical composition of PM10 strongly differs from one location to another. PM10 chemistry is different from PM2.5 and the associations reported here are only valid for PM10. Some components that might mainly reside in the coarse mode are positive factors in the multiple linear regression models (e.g Ti in OP ESRv). They can display a different final health impact, since a fraction of PM10 does not penetrate all the way

to lung. Also, the results of the ESR assay warrant caution due to our back correction of the ESR signal linked to the non linear response of the assay. Finally, multiple model result for the GSH assay is to be considered with caution since normal distribution was not reached in the first step of the analysis. Finally, these analyses are only relevant for PM10 when some health studies are now taking PM2.5 into account. Additional studies addressing comparison of OP results associated with PM10 and PM2.5 are needed (Gali et al., 2017; Styszko et al., 2017).

Minor comments

5) It appears both ASC and AA are used as abbreviation to refer to ascorbic acid. Why two abbrevations?

It's true that both ASC and AA are abbreviations for ascorbic acid. These distinct abbreviations were used because of the two tests using acid ascorbic and that also use different analyses techniques. In the case of the AA: the test consists in measuring the depletion of a single antioxidant (ascorbic acid) with spectrophotometry techniques. For the ASC, the depletion of ascorbic acid is measured with HPLC system. ASC is part of the RTLF assay in which the depletion of three antioxidants is measured (ascorbic acid, glutathione, and urate).

6) The reference "Chevrier 2016" is given in French. Please provide an English translation and also how this reference can be accessed.

This reference is a PhD manuscript. The English translation was realized:

"Wood heating and air quality in the Arve Valley : definition of a surveillance system and impact of a renovation policy of old devices"

7) Please define "DPCC". The first appearance is line 6 on page 4.

We added the word associated to DPPC:

Page 4 line 10 : For extraction procedure, PM samples were extracted in using a

Gamble + DPPC (dipalmitoylphosphatidylcholine) solution and vortexed at maximum speed during 2h at 37°C (Calas et al., 2017).

8) Page 5, line 9: Is "the DDT assay" supposed to be "the AA assay" instead?

The sentence was correct. However to make it more clear, the sentence was modified as follow:

Page 5 line 14 : A semi-automated procedure using the same plate reader than for the DTT assay was applied using Greiner UV-Star® 96 well plates, this assay was based on the modified assay from Zielinski et al. (1999) and Mudway et al. (2004).

9) Figure S3: is the y-axis label supposed to be "nmol AA/min"?

Yes, thank you for that, the y-axis label has been modified in the latest version of the SI

10) Page 11, lines 2-3: please cite a reference for the criterion for determining whether a correlation is strong or moderate.

This criterion was arbitrary chosen, however it is in the range of criterion commonly found in the literature. For example, in the study of Yang et al. (2014): moderate correlation criteria was attributed to spearman's correlation (rs) ranged between rs = 0.61 - 0.68. In the same study, very high correlations were attributed to rs > 0.90 and high correlation for rs = 0.86 – 0.96.In the study of Janssen et al. 2014: high correlations were attributed to rs ranged between 0.77 and 0.96. Lower correlation were attributed to rs ranged between 0.4 – 0.6. They also reported the criteria "moderate" for rs = 0.39 – 0.62 .

We added more about that in the following sentence;

Page 11 line 2: Spearman correlations (rs) between the assays were calculated over the sampling year. Correlations were considered as strong for rs > 0.70 and moderate for rs (> 0.45 and <0.70) which are commonly criteria that can be found in the literature (Janssen et al., 2014; Yang et al., 2014);

11) Table S4: one entry of Mg –> Mg2+; NO3–> NO3-; NH4–>NH4+

Thanks again for this comment; the modifications have been taken into account in the latest version.

| Months | n | | | | |
|--------|-----|-----|-----|-----|-----|
| | DTT | AA | ESR | GSH | ASC |
| Nov | 7 | 7 | 7 | 7 | 7 |
| Dec | 10 | 10 | 10 | 10 | 8 |
| Jan | 11 | 11 | 11 | 11 | 10 |
| Feb | 7 | 7 | 7 | 7 | 7 |
| Mar | 8 | 8 | 8 | 8 | 8 |
| Apr | 9 | 9 | 7 | 9 | 9 |
| May | 5 | 5 | 4 | 5 | 5 |
| Jun | 8 | 8 | 5 | 8 | 8 |
| Jul | 5 | 5 | 3 | 5 | 5 |
| Aug | 8 | 8 | 3 | 8 | 8 |
| Sep | 9 | 9 | 4 | 9 | 9 |
| Oct | 11 | 11 | 6 | 11 | 11 |
| **Total** | **98** | **98** | **75** | **98** | **95** |

**Fig. 1.**

---

## Author Response (AR2)

**Co-Editor Decision: Publish subject to minor revisions (review by editor)** (29 Apr 2018) by Nga Lee Ng
Comments to the Author:
Dear authors,

Thank you for the detailed response to reviewers' comments. I think you have addressed their comments sufficiently. I just have two minor comments. The manuscript will be accepted for publication in ACP once these are addressed.

1. A number of recent studies have indicated that oxidative activities from different PM components are not always additive (e.g., Tuet et al., 2016 AE, Xiong et al., 2017 ES&T, Wang et al., 2017 ACP, Yu et al., 2018 ES&T, Tuet et al., 2018 Sci. Rep., etc). It would be useful to briefly discuss the potential limitations of the linear regression model as well (page 15).

We included this comment in the limitation section as following:

Several limitations can be attributed to this study. Most important, all of these results have been obtained for a specific location and cannot be generalized as chemical composition of $PM_{10}$ strongly differs from one location to another. $PM_{10}$ chemistry is different from $PM_{2.5}$ and the associations reported here are only valid for $PM_{10}$. Some components that might mainly reside in the coarse mode are positive factors in the multiple linear regression models (e.g Ti in OP ESRv). They can display a different final health impact, since a fraction of $PM_{10}$ does not penetrate all the way to lung. Also, the results of the ESR assay warrant caution due to our back correction of the ESR signal linked to the non-linear response of the assay. Finally, multiple model result for the GSH assay is to be considered with caution since normal distribution was not reached in the first step of the analysis. Moreover, multiple non-linear regression models should also be investigated since several studies have shown that oxidative potentials from different PM components are not always additive (Tuet et al., 2016; Tuet et al., 2018; Wang et al., 2017; Xiong et al., 2017; Yu et al., 2018). Finally, these analyses are only relevant for $PM_{10}$ when some health studies are now taking $PM_{2.5}$ into account. Additional studies addressing comparison of OP results associated with $PM_{10}$ and $PM_{2.5}$ are needed (Gali et al., 2017; Styszko et al., 2017).

2. Regarding metal-organic binding interactions (page 15, line 15), it would also be appropriate to cite Verma et al., 2012 ES&T and Tuet et al., 2016 AE.

Added accordingly page 15, line 15.

Best,
Sally

Thank you for your useful comments,

On behalf all co-authors, Gaëlle Uzu